# Short- and Long-Term Effects of Rehabilitation after Perimesencephalic Subarachnoid Hemorrhage

**DOI:** 10.3390/diseases9040069

**Published:** 2021-10-07

**Authors:** Jens Schmitz, Sepide Kashefiolasl, Nina Brawanski, Nazife Dinc, Florian Gessler, Christian Senft, Stephanie Tritt, Volker Seifert, Jürgen Konczalla

**Affiliations:** 1Department of Neurosurgery, Goethe-University Hospital, 60528 Frankfurt am Main, Germany; Sepide.Kashefi@gmx.de (S.K.); Nina.brawanski@kgu.de (N.B.); Dinc.nazife@gmail.com (N.D.); Flo.gessler@gmail.com (F.G.); c.senft@med.uni-frankfurt.de (C.S.); v.seifert@em.uni-frankfurt.de (V.S.); J.Konczalla@med.uni-frankfurt.de (J.K.); 2BG Klinikum Duisburg gGmbH, 47249 Duisburg, Germany; 3Institute of Neuroradiology, HELIOS HSK Wiesbaden, 65199 Wiesbaden, Germany; stephanie.tritt@gmx.de; 4Institute of Neuroradiology, Goethe-University Hospital, 60528 Frankfurt am Main, Germany

**Keywords:** non-aneurysmal, aneurysm, perimesencephalic, non-perimesencephalic, prepontine, subarachnoid hemorrhage, SAH, short-term, long-term outcome, SF-36

## Abstract

In about 25% of patients with spontaneous subarachnoid hemorrhage (SAH), a bleeding source cannot be identified during radiological diagnostics. Generally, the outcome of perimesencephalic or prepontine (PM) SAH is known to be significantly better than after non-PM SAH. Data about long-term follow-up concerning physical and mental health are scarce, so this study is reports on long-term results. We measured the influence of PM SAH on a quality-of-life modified Rankin (mRs) scale after six months. For long-term follow-up, a SF-36 questionnaire was used. Questionnaires were sent out between 18 and 168 months after ictus. In 37 patients, a long-term follow-up was available (up to 14 years after SAH). Data detected with the SF-36 questionnaire are compared to reference applicability to the standard population. In total, 37 patients were included for further analysis and divided in 2 subgroups; 13 patients (35%) received subsequent rehabilitation after clinical stay and 24 (65%) did not. In the short-term outcome, a significant improvement from discharge until follow-up was identified in patients with subsequent rehabilitation, but not in the matched pair group without rehabilitation. When PM SAH was compared to the standard population, a reduction in quality of life was identified in physical items (role limitations because of physical health problems, physical functioning) as well as in psychological items (role limitations because of emotional problems). Subsequent rehabilitation on PM SAH patients probably leads to an increase in independence and better mRs. While better mRs was shown at discharge in patients without subsequent rehabilitation, the mRs of rehabilitants was nearly identical after rehabilitation. Patients with good mRs also reached high levels of health-related quality of life (HRQoL) without rehabilitation. Thus, subsequent rehabilitation needs to be encouraged on an individual basis. Indication criteria for subsequent rehabilitation should be defined in further studies to improve patient treatment and efficiency in health care.

## 1. Introduction

Spontaneous subarachnoid hemorrhage (SAH) is usually caused by an aneurysm rupture of intracranial arteries vessels often associated with hypertensive blood pressure or innate malformations. SAH is associated with morbidity caused by tissue damage due to bleeding and secondary tissue damage caused by complex neuroinflammatory pathways [1].

However, the rate of non-aneurysmal SAH (naSAH) has increased in recent decades. Currently, in about 25% of patients with spontaneous SAH, a bleeding source cannot be identified by radiological diagnostics [2,3]. An anatomical classification differentiates between perimesencephalic or prepontine (PM) and non-perimesencephalic (NPM) hemorrhage localization [4]. Generally, the outcome of PM SAH is known to be significantly better than after NPM SAH [5,6,7].

Nonetheless, data about short-term and long-term follow-up due to physical and mental health are scarce; thus, this study reports these results [8]. Rehabilitation was identified as a positive prognostic outcome factor for NPM SAH [9]; therefore, we used the modified Rankin scale (mRs) for short-term follow-up after six months and the SF-36 questionnaire to evaluate whether rehabilitation had an effect on long-term outcomes after PM SAH.

## 2. Materials and Methods

From 1999 to 2012, 1404 patients with spontaneous, non-traumatic SAH were treated at our institution. A prospective long-term outcome evaluation was performed in patients with non-aneurysmal SAH. In total, 173 patients had a non-aneurysmal SAH. Furthermore, 87 patients had a perimesencephalic SAH and, in 37 patients, a long-term follow-up was available (Figure 1) [10].

Study inclusion and exclusion criteria:

Inclusion: PM SAH after complete work-up; at least 18 months after PM SAH;

Exclusion: aneurysmal SAH, non-PM SAH, non-responding; withdrawal from study (long-term SF-36).

### 2.1. Diagnostic Work-Up and SF-36 Questionnaire

Diagnostics were proceeded in determined algorithms. For securing the diagnosis of SAH, computed tomography (CT) or lumbar puncture was used. For a definite exclusion of aneurysmal SAH or intracerebral hemorrhage, a digital subtraction angiography (DSA) was used. Additionally, magnetic resonance imaging (MRI) scanning of the whole central nervous system was used to exclude other bleeding sources [2,11,12].

Definition of perimesencephalic SAH was accomplished by van Gijn et al. and Rinkel et al. [4,13].

Short-term follow-up after six months was evaluated by a modified Rankin scale (mRs). MRs is a standardized tool for disability after suffering events with neurological affections. The scale runs from zero to six in which zero is treated as equivalent to no symptoms (1.no significant disability, 2. slight disability, 3. moderate disability, 4. moderately severe disability, 5. severe disability). The worst outcome (dead) is figured with six [14].

A short-form-36 (SF-36) questionnaire was sent out between 18 and 168 months after ictus.

For measuring the influence of PM SAH on quality of life, a prospective analysis using the SF-36questionnaire was used. The SF-36 questionnaire is a standardized measuring tool that allows one to evaluate the health-based physical and psychological quality of life in 36 questions. Results are divided into two main sectors with four subcategories each. Psychological health reflects the effects of health due to perceived mental mood. Physical health shows the impact on health-related quality of life (HQRL) due to resilience and functioning.

Vitality (four questions), social functioning (two questions), role limitations due to emotional problems (three questions) and general mental health (five questions) comprise psychological health.

Physical functioning (ten questions), role limitations due to physical health problems (four questions), bodily pain (two questions) and general health problems (five questions) comprise physical health [15,16]. General health perception cannot be assigned to psychological or physical health (one question) (Table 1).

Results are scored on a scale from 0 to 100, in which 100 shows a maximum health-related quality of life [15,16].

Data obtained with the SF-36 questionnaire were recorded and analyzed with SPSS (SPSS Inc., IBM Corp., Ehningen, Germany, accessed date: 20 April 2016) and compared to a reference population. Therefore, SPSS data from a US health survey (n = 2474) were used. The SF-36 questionnaires were sent via mail. If we had not received an answer within three months, telephone interviews were conducted.

For statistical analyses the Mann–Whitney U-test was used. For categorical variables, the Fisher’s exact test was used. Every *p*-value < 0.05 was considered statistically significant.

### 2.2. Trial Registration

This prospective, non-interventional evaluation was permitted by the ethics committee of Goethe University Hospital, Frankfurt am Main. This study is registered with www.clinicaltrials.gov (accessed on 8 January 2015), identifier No.: NCT02334657.

### 2.3. Rehabilitation

After initial hospitalized acute treatment, subsequent rehabilitation in a specialized neurological rehabilitation establishment was offered. The choice of patients for subsequent rehabilitation was made according to the patient’s clinical situation and their agreement. During rehabilitation, treatment patients received varying, patient-adapted therapies. These were physical and ergo therapy, specific training to manage everyday situations, nutrition guidance and psychological therapy. On average, rehabilitation took three to four weeks and started immediately after clinical treatment.

## 3. Results

### 3.1. Patient Collective

In total, 87 patients met the criteria for PM SAH. Until long-term follow-up, 12 patients died (14%), 37 patients declined the follow-up and 1 patient could not be informed. Therefore, 37 patients were included for further analysis. Questionnaires were answered after 77 months post-SAH in median (range 33–159 month). Median age of patients at ictus of PM SAH was 55.1 years (±11.5). From 37 patients, 13 patients (35%) received subsequent rehabilitation and 24 (65%) did not.

### 3.2. Short-Term Outcome after 6 Months

Overall, the mean mRS of all patients was 1.56 (Table 2). Patients who received rehabilitation showed a worse mRS than the collective with rehabilitative treatment (mRs 1.7; ±0.5) at discharge. Six months after ictus and rehabilitation, mRS increased to 0.8. Those patients who did not receive rehabilitation showed an mRS of 1.36 (±0.74), better than the collective, at discharge. Overall improvement since discharge was 0.6 (±0.59) on the mRS scale. Patients with subsequent rehabilitation showed a worse mRs in the six-month follow-up (mRs 0.8; ±0.6) than patients without subsequent rehabilitation (0.54; ±0.59). A statistically significant improvement from discharge until follow-up was only identified in patients with subsequent rehabilitation (*p* < 0.001; Table 2).

### 3.3. Long-Term Outcome of PM SAH and Comparison with Standard Population

When PM SAH Was compared to the standard population, a reduction in quality of life in every field of SF-36 was identified. In social functioning, HRQoL nearly reaches the standard population. Differences in physical pain, general health problems, vitality and general mental health were lower. Higher deviations in the reduction in HRQoL were shown in physical functioning, role limitations due to physical health problems and role limitations due to emotional problems. The only statistically significant reductions in HRQoL were revealed in general health problems, role limitations due to emotional problems and role limitations due to emotional problems (*p* < 0.05; Figure 2).

### 3.4. Long-Term Outcome of PM SAH Patients with Subsequent Rehabilitation

The comparison between the results of PM SAH patients with subsequent rehabilitation and the standard population shows impairments in all fields. Lower differences can be seen in physical pain, vitality, social functioning and general mental health. Higher reductions are shown in physical functioning, role limitations due to physical problems, general health problems and role limitations due to emotional problems. Reductions in HRQol with statistically significant relevance are only shown in general health problems and role limitations due to emotional problems (*p* < 0.05; Figure 3.)

The HRQoL of patients suffering with PM SAH with subsequent rehabilitation was compared to patients suffering with PM SAH without rehabilitation. The rehabilitation group shows a worse HRQoL in physical functioning, general health problems and role limitations due to emotional problems. A clear improvement can be seen in role limitations due to physical health problems and vitality.

## 4. Discussion

Non-aneurysmal SAH, especially PM SAH, was associated with favorable outcomes, but data related to long-term outcomes are scarce [8,9,10,17,18]. In general, this study shows that non-aneurysmal SAH relates to a reduction in quality of life in a physical and psychological manner. Patients with NPM SAH had a worse outcome than patients suffering with PM SAH [10]. Nonetheless, some patients with PM SAH also showed some benefits from undergoing rehabilitation. Data referring to the influence of subsequent rehabilitation after clinical stay are rare.

### 4.1. Outcome at Discharge until 6 Months Short-Term Follow-Up

At the point of dismissal of clinical stay, both subgroups (patients with or without rehabilitation) showed a decrease in health-related quality of life and the ability to manage everyday life. Patients who received subsequent rehabilitation showed a worse functional outcome (mRs 1.7 ± 0.5) at discharge than patients who did not use subsequent rehabilitation (mRs 1.36 ± 0.74) (Table 2) [14,19].

In our six-month follow-up, patients with subsequent rehabilitation improved mRS (mRs 0.8 ± 0.6), and patients without subsequent rehabilitation also improved mRs (mRs 0.54 ± 0.59). Here, a direct positive influence of subsequent rehabilitation cannot be proved because both subgroups improved mRS during the six-month follow-up. Despite its significantly worse starting situation at clinical discharge, the statistically significant increase in mRs in the subsequent rehabilitation group is most likely affiliated with subsequent rehabilitation. Patients with slightly reduced parameters at discharge achieved acceptable results without rehabilitation. Certainly, the generally positive influence of rehabilitation assumes that subsequent rehabilitation in these cases will lead to a significantly better outcome.

### 4.2. PM SAH at Long-Term Follow-Up

Suffering PM SAH leads to reduction in HRQoL in all items of SF-36 compared to the normal population. In this study, only three items are statistically significant (physical functioning, role limitations due to physical problems, role limitations due to emotional problems). The reduction in psychological role function is probably caused by the partial loss of physical role function. The tendential reductions, without being statistically significant within the remaining areas, show that, despite having adequate medical treatment and subsequent rehabilitation, a slight reduction in HRQoL remains. Even though there are fewer items with statistical significance, the trend is apparent.

Comparing patients who underwent and did not undergo subsequent rehabilitation to a standard population HRQoL, the rehabilitation subgroup showed less HRQoL in physical functioning, general health problems and role limitations due to emotional problems, but better HRQoL in role limitations due to physical health problems and vitality. Reductions in these items might be explained by general health issues that are connected to SAH, as well as knowledge patients’ own health-related problems. Nevertheless, the fewer reductions in role limitations due to physical health problems and vitality might be enhanced by subsequent rehabilitation. Subsequent rehabilitation also comprises the initiation of adequate medical therapy. Severe headache is highly associated with SAH and is known as an important parameter for reducing HRQoL [20]. The low difference in pain between patients with subsequent rehabilitation and the standard population might be explained by adequate medical therapy initiated during rehabilitation.

Interestingly, only one physical and psychological item was significantly reduced in patients with subsequent rehabilitation. Therefore, it seems that patients with PM SAH improved from rehabilitation until short-term follow-up, and that this improvement lasted until long-term follow-up. This is particularly important given that a worse discharge outcome is known to be an independent predictor for quality of life, as measured by the SF-36 in stroke patients [10,19].

### 4.3. Limitations and Generalizability

This study has several limitations. First, statistical analysis was performed in a single center. In addition, the study design is partly retrospective. Moreover, some patients could not be determined and long-term follow-up was only available in 37 patients. Nevertheless, a response of 51% is good to acceptable for a period of 15 years.

The total number of participants is low but, given the the low prevalence of PM, SAH is still high. However, even the lower total number is useful for showing trends, which was the purpose of this study.

It is meaningful to carry on research in this area to gain more specific knowledge about the effects of PM SAH on health and health-related quality of life. In this sense, treatment and rehabilitation concepts can be improved for the benefit of patients.

## 5. Conclusions

Patients suffering PM SAH regularly have a favorable outcome and can live independently. However, we identified a reduction in quality of life by SF-36 long-term evaluation. This was applicable to patients who received subsequent rehabilitation and for those who did not. Patients with PM SAH had reduced HRQoL.

Subsequent rehabilitation on PM SAH patients probably leads to an increase in independence and better mRs. While better mRs was shown at discharge in patients without subsequent rehabilitation, the mRs of rehabilitants was nearly identical.

Patients with PM SAH benefit from subsequent rehabilitation with improved functioning in everyday life, in short-term follow-up, and generally benefit in terms of health-related quality of life in relation to long-term outcomes. Patients with good mRs also reach high HRQoL without rehabilitation. Thus, subsequent rehabilitation needs to be encouraged on an individual basis. If mRs is acceptable at discharge, patients suffering from PM SAH will probably obtain a good HRQoL in the long-term follow-up. However, if mRS at discharge is worse, subsequent rehabilitation will nearly effect the same HRQoL. Indication criteria for subsequent rehabilitation should be defined in further studies to improve patient treatment and efficiency in health care.

## Figures and Tables

**Figure 1 diseases-09-00069-f001:**
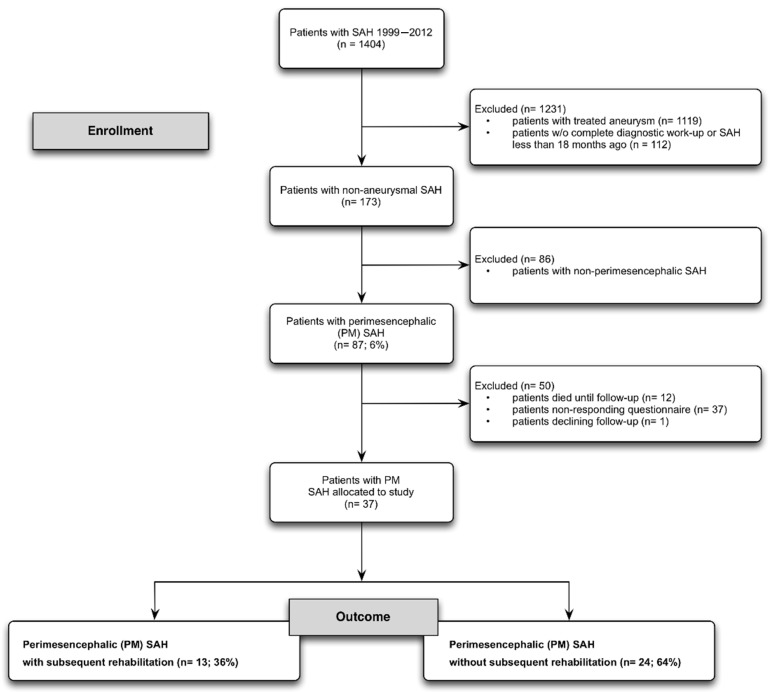
Flow diagram with detailed information about enrollment and long-term follow-up. From over 1400 patients with SAH, 87 patients with PM SAH were identified. Overall, a good response rate was achieved, especially against the background of patients suffering from NASAH starting 20 years ago.

**Figure 2 diseases-09-00069-f002:**
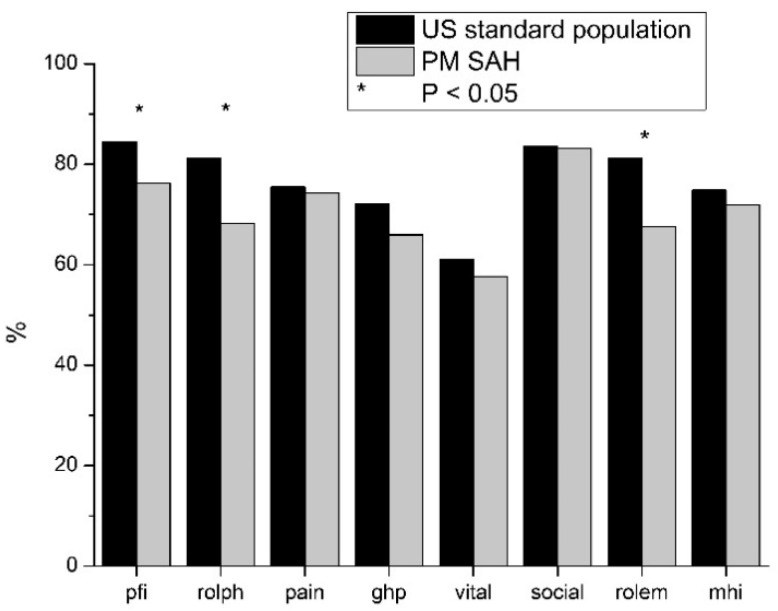
Comparison of outcome of patients with PM SAH and standard population. Long-term outcome compared to standard population, measured by SF-36. On the x-axis, the eight items of the SF-36 are shown, and the y-axis is scaled metrically. Standard population (*black*) was compared with PM SAH (*grey*). At long-term outcome, Pfi, rolph and rolem are significantly reduced after PM SAH compared to the standard population.

**Figure 3 diseases-09-00069-f003:**
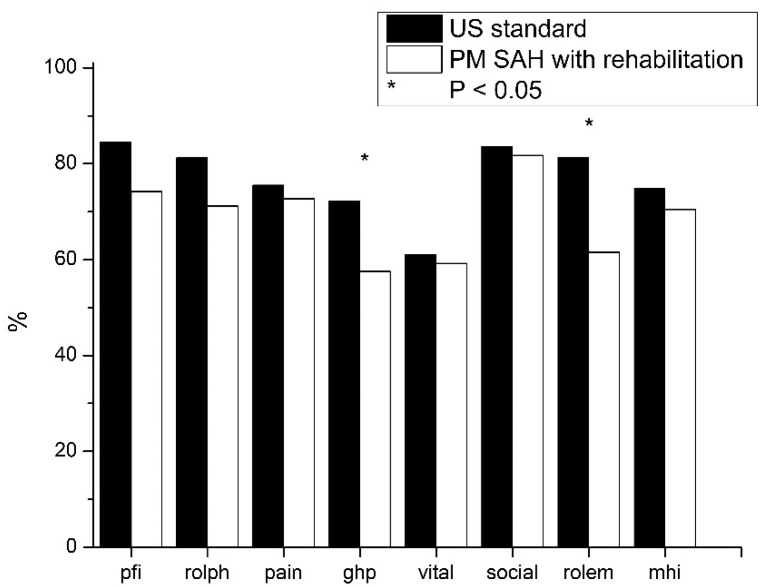
Outcome of patients with PM SAH and subsequent rehabilitation compared to standard population. In patients with PM SAH and rehabilitation (*white*) group, only ghp and rolem are significantly reduced compared to the standard population (*black*).

**Table 1 diseases-09-00069-t001:** The items of SF-36 questionnaire.

	Item	Abbreviation
**Pyhsical**	Physical functioning	pfi
	Role limitations due to physical health problems	rolph
	Physical pain	pain
	General health problems	ghp
**Psychological**	Vitality	vital
	Social functioning	social
	Role limitations due to emotional problems	rolem
	General mental health	mhi

**Table 2 diseases-09-00069-t002:** Distribution of recirculated questionnaires and short-term outcome. Shown are patients’ age and outcome stratified by different subgroups: all PM SAH patients (n = 37), patients without subsequent rehabilitation (n = 24), patients with subsequent rehabilitation (n = 13) and a matched pair group of patients without subsequent rehabilitation (n = 13). To minimize the influence of selection, a matching procedure was necessary. WFNS grade and age were used as matching parameters.

Characteristics	All PM SAH	Without Subsequent Rehabilitation	With Subsequent Rehabilitation	*p* (with vs. without Rehabilitation)
No. of patients	37 (100%)	24 (65%)	13 (35%)	NS
mean age ± SD	55.1 ± 9.6	53.7 ± 10.8	57.8 ± 6.2	NS
Outcome at discharge (mean mRS ± SD)	1.56 ± 0.64	1.36 ± 0.74	1.7 ± 0.5	NS
short-term outcome (mean mRS ± SD)	0.6 ± 0.59	0.54 ± 0.59	0.8 ± 0.6	NS
Improvement from discharge to short-term outcome (6 months)	0.69	0.82	0.9	NS
*p* (mRS discharge vs. mRS 6 months FU)	NS	NS	<0.001	

PM SAH, perimesencephalic subarachnoid hemorrhage; NS, not significant (*p* > 0.05); SD, standard deviation; mRS, modified Rankin scale (mRS); FU, follow-up.

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
