# Peer review of "Short- and Long-Term Effects of Rehabilitation after Perimesencephalic Subarachnoid Hemorrhage"

_diseases, 2021, doi:10.3390/diseases9040069_

Round 1

Reviewer 1 Report

Clinical outcome of subarachnoid haemorrhage ranges from death to (near-)complete recovery. Rehabilitation is extensively used with the aim to improve outcome and physical, emotional functionality. However, the efficacy of rehabilitation is highly variable, with often no clear explanation. The Authors assessed different groous of SAH patients and their physical and mental condition (up to 14 years follow-up) with and without rehabilitation. The results demonstrate that rehabilitation is of not expected to be quite efficient and provides indications in which patient population is the rehabilitation likely efficient. This further supports the view that it shall be conducted and designed individually and warrants further studies in the field. 

The strength and limitations of the study are clearly presented and discusssed. Given the importance of the topic it is an important contribution to the field.

Specific comments:

line 17: add abbreviation mRs

line 30 specify abbreviation HRQoL

line 87: space missing between words

use p in italics and non-capital letter throughout the text including legends

line 172: PM SAH (use abbreviation consistently; in some pleace it reads PM- SAH)

line 179: specify non-aSAH (non-aneurysmal SAH ?)

Author Response

Response to Reviewer 1 Comments:

Dear Reviewer 1,

Thank you for the rapid review and your comments.

All the noted mistakes were adjustet.

Reviewer 2 Report

This paper nicely summarizes the role of rehabilitation in perimesencephalic SAH. The patients needing rehab often had worse mRS scores but subsequently became comparable on extended follow up outcomes. The benefit is that rehab might be strongly suitable for this population but further prospective studies are warranted. The authors clearly state the limitations. 

The study would benefit from the following.

Extended background into the mechanisms of SAH should be included in the background. Emphasis on neuroinflammation should be addressed. The paper is lacking a key reference PMID: 27049383.

Furthermore, headaches can effect quality of life in this patient population. This should be addressed in the discussion. The paper is lacking a key reference PMID: 33771620.

If these points are adequately addressed and expanded, the paper would be of interest to the readership. 

Author Response

Response to Reviewer 1 Comments

Dear Reviewer,

Thank you for the contribution of further interesting subjects to this manuscript with definite interest to the readership.  I hope all the proceeded changes address the issues of neuroinflammatory response and headache. I have to apologize the superficiality of the headache issue but more detailed information is not pictured in our study.

This manuscript is a resubmission of an earlier submission. The following is a list of the peer review reports and author responses from that submission.